# Intra-Modal Proxy Learning for Zero-Shot Visual Categorization with CLIP

**Qi Qian**[1*]    **Yuanhong Xu**[2]    **Juhua Hu**[3]
[1] Alibaba Group, Bellevue, WA 98004, USA
[2] Alibaba Group, Hangzhou, China
[3] School of Engineering and Technology, University of Washington, Tacoma, WA 98402, USA
`{qi.qian, yuanhong.xuyh}@alibaba-inc.com, juhuah@uw.edu`

## Abstract

Vision-language pre-training methods, e.g., CLIP, demonstrate an impressive zero-shot performance on visual categorizations with the class proxy from the text embedding of the class name. However, the modality gap between the text and vision space can result in a sub-optimal performance. We theoretically show that the gap cannot be reduced sufficiently by minimizing the contrastive loss in CLIP and the optimal proxy for vision tasks may reside only in the vision space. Therefore, given unlabeled target vision data, we propose to learn the vision proxy directly with the help from the text proxy for zero-shot transfer. Moreover, according to our theoretical analysis, strategies are developed to further refine the pseudo label obtained by the text proxy to facilitate the intra-modal proxy learning (InMaP) for vision. Experiments on extensive downstream tasks confirm the effectiveness and efficiency of our proposal. Concretely, InMaP can obtain the vision proxy within one minute on a single GPU while improving the zero-shot accuracy from $77.02\%$ to $80.21\%$ on ImageNet with ViT-L/14@336 pre-trained by CLIP.

## 1 Introduction

Vision-language pre-training in CLIP aims to align image-text pairs to facilitate multi-modal understanding for downstream tasks [26]. By optimizing the contrastive loss consisting of images and their corresponding text descriptions, CLIP demonstrates an impressive zero-shot transfer of the pre-trained model to generic tasks. Concretely, given a downstream data set, each class has a proxy obtained from the text embedding about the corresponding class name, and then images can be categorized by finding the nearest class proxy. With class names only, zero-shot accuracy on ImageNet [28] can achieve $77.02\%$ with ViT-L/14@336 [9] pre-trained by CLIP.

After the success of CLIP, many research efforts have been devoted to improving the transfer performance on downstream tasks. Most of developed methods can be cast into two categories, that is, few-shot refinement and zero-shot enhancement. Few-shot methods require a small amount of labeled data from the target task to refine the prediction [12, 41, 42]. While these methods show a better performance than zero-shot learning, labeled examples from the target domain may not be available in real-world applications. Therefore, many methods focus on improving the zero-shot learning itself by exploring side information. On one hand, many pre-trained large language models (LLMs), e.g., GPT-3 [4], can be leveraged to obtain better text proxy for classification [20]. On the other hand, unlabeled data from the target domain can be adopted to fine-tune an adapter network or input prompts [23, 30] for alignment. In this work, we focus on understanding the behavior of CLIP by exploiting the unlabeled data without external LLMs or additional architectures for zero-shot categorization.

---

*Corresponding author

37th Conference on Neural Information Processing Systems (NeurIPS 2023).

When considering pre-trained CLIP models and excluding any auxiliary networks for zero-shot transfer, many existing methods try to optimize input text prompts to align images and the text of class names better [30, 33, 42]. However, recent observations demonstrate that the modality gap between text (i.e., language) and vision is still significant after the optimization of CLIP [18]. Consequently, the efforts on improving the text proxy can be ineffective for the vision task.

To mitigate the problem from the gap between the text and vision space, we propose to obtain proxies of classes in the vision space directly. First, our theoretical analysis in Proposition 1 shows that the modality gap will be preserved due to a small temperature parameter in the contrastive loss of CLIP, which confirms the empirical observation in [18]. Furthermore, Proposition 2 indicates that the optimal proxy for the vision task can only be obtained from the vision space. Therefore, with the gap between the text and vision space, directly using the text proxy makes the zero-shot transfer sub-optimal. To recover the optimal vision proxy, we develop an intra-modal proxy learning (InMaP) method to obtain visual proxies with pseudo labels from text proxies. The proposed method is illustrated in Fig. 1. With fixed features extracted by CLIP encoders, the optimization problem in InMaP is convex without encoders in the loop, which can be solved efficiently with gradient descent [3]. Our main contributions can be summarized as follows.

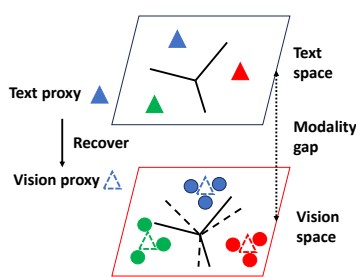

Figure 1: Illustration of our proposed intra-modal proxy learning (InMaP). Triangles denote the class proxy while circles denote the vision data. Different classes are denoted with different colors. By recovering the vision proxy, it can align with vision data better (i.e., from solid line as in the text space to dashed line in the vision space).

- We theoretically show that the modality gap cannot be reduced sufficiently by minimizing the contrastive loss in CLIP. While the optimal proxy for image classification is in the vision space, the gap degenerates the zero-shot performance of the text proxy as shown in Theorem 1.

- We propose a novel algorithm to obtain the intra-modal proxy directly from the vision space. By recovering the vision proxy with help from the text proxy and an appropriate temperature suggested by Proposition 3 for calibration, effective proxies can be observed from unlabeled data.

- Our Theorem 2 indicates that the accuracy of pseudo labels from the text proxy is essential to recover the proxy in the vision space. Therefore, we further develop strategies to refine pseudo labels according to the data distribution by optimal transport (OT) [34].

- Experiments on 14 downstream vision tasks demonstrate both the effectiveness and efficiency of our proposal. Concretely, InMaP can achieve $80.21\%$ accuracy on ImageNet with ViT-L/14@336 using a single GPU within one minute.

## 2    Related Work

Before CLIP, zero-shot learning that aims to identify examples of novel classes without any labeled training data has attracted much attention since [22]. However, most of existing works are only able to discover new classes closely related to the training classes [1, 5, 6, 11, 22, 38], where they share the similar attributes, and have to train an individual model for each task. On the contrary, pre-training on large-scale data with the contrastive loss aligns visual and language features and makes a single CLIP model applicable for diverse downstream tasks in a straightforward way. Compared with conventional zero-shot methods, the pre-training data in CLIP may be overlapped with downstream tasks, which can result in data leak for evaluation. While the issue can be addressed by eliminating the overlapping data, the influence on the performance is negligible as discussed in [26].

After pre-training with image-text pairs, the obtained CLIP models can be transferred to various downstream tasks with few-shot or zero-shot strategies. We briefly review these two related directions as follows.

**Few-shot transfer**    CLIP demonstrates that an ensemble of input text prompts for the text encoder can improve the zero-shot performance by a significant margin. Therefore, many methods try to

further refine the prompts. Given a small set of labeled data from the target vision tasks, CoOp [42] optimizes input prompts for the text encoder with labels. By formulating the prompts as learnable variables, it can obtain task-dependent prompts without any hand-crafted designs. On the contrary, CLIP-Adapter [12] optimizes obtained output features from both encoders. However, it adds additional bottleneck layers for training, which increases the complexity of optimization. To reduce the efforts of training with encoders, Tip-Adapter [41] demonstrates a training-free method that incorporates the prediction from a cache model consisting of the labeled image data and that from the text proxy. In this work, we also consider to minimize the training efforts by excluding encoders in optimization and the intra-modal proxy can be obtained from extracted features.

**Zero-shot transfer** Since labeled data are not always available, many efforts are devoted to improving zero-shot transfer. Some methods explore other large pre-trained models as the supplementary for CLIP. VisDesc [20] leverages GPT-3 to generate rich context descriptions for given class names, and thus shows superior performance over the simple prompt in CLIP. SuS-X [33] includes additional images to mitigate the modality gap. Concretely, auxiliary images can be obtained either from a large data set (e.g., LAION [29]) or a text-to-image generation model (e.g., stable diffusion [27]). The additional large models will inevitably increase the inference time, hence, recent methods consider to leverage the unlabeled target data for fine-tuning. UPL [15] and TPT [30] utilize the unlabeled data to optimize learnable input text prompts. SVL-Adapter [23] trains an additional encoder from the unlabeled data with self-supervised learning for ensemble. Unlike existing methods, our proposal does not need any additional model and we learn the proxy for each class from the vision space directly.

## 3 The Proposed Method

In this section, we first analyze the modality gap in CLIP, and then elaborate the proposed method that recovers the proxy in the vision space to reduce the influence from the modality gap.

### 3.1 Cross-Modal Learning in CLIP

By pre-training paired images and text, a vision-language model, e.g., CLIP [26], is capable of classifying unlabeled images in the zero-shot manner. Let $f(\cdot)$ and $g(\cdot)$ denote the encoder for images and text in the pre-trained CLIP model, respectively. Given a set of unlabeled image data $\{x_i\}_{i=1}^n$ and the unique class names for the set $\{z_j\}_{j=1}^C$, their corresponding visual and text representations can be extracted as

$$\mathbf{x}_i = f(x_i); \quad \mathbf{z}_j = g(z_j)$$

where $\mathbf{x}_i \in \mathcal{R}^d$ and $\mathbf{z}_j \in \mathcal{R}^d$ have the same dimension and a unit norm. $d$ is the dimension of features. $\mathbf{z}_j$ can be considered as the proxy for the $j$-th class.

With the features of the $i$-th image and all text proxies, the image can be classified as

$$y_i = \arg\max_j \mathbf{x}_i^\top \mathbf{z}_j$$

where $y_i$ is the prediction from the zero-shot classification. The capacity of zero-shot prediction is from the alignment between vision and language that is optimized by the pre-training objective in CLIP. Given image-text pairs as $(x_i, t_i)$, two encoders are learned by minimizing the contrastive loss

$$\sum_i -\log \frac{\exp(\mathbf{x}_i^\top \mathbf{t}_i/\tau)}{\sum_j^m \exp(\mathbf{x}_i^\top \mathbf{t}_j/\tau)} - \log \frac{\exp(\mathbf{t}_i^\top \mathbf{x}_i/\tau)}{\sum_j^m \exp(\mathbf{t}_i^\top \mathbf{x}_j/\tau)}$$

where $m$ is the mini-batch size, $\mathbf{t}_i = g(t_i)$ is the extracted text representation of $t_i$ with a unit norm, and $\tau$ is the temperature for the normalized cross entropy loss.

While CLIP demonstrates an impressive zero-shot performance, the cross entropy loss focuses on cross-modal ranking and can be sub-optimal for intra-modal vision categorization. It is because that cross entropy loss is closely related to a triplet loss [25] as

$$\forall j, \quad \mathbf{x}_i^\top \mathbf{t}_i - \mathbf{x}_i^\top \mathbf{t}_j \geq 0; \quad \mathbf{t}_i^\top \mathbf{x}_i - \mathbf{t}_i^\top \mathbf{x}_j \geq 0$$

which mainly aims to rank the multi-modal positive pair over negative pairs rather than to mix the text and vision space for modality gap reduction. Moreover, a small temperature, e.g., 0.01 in CLIP, will

amplify the difference between pairs and preserve the distance between modalities. The phenomenon can be stated as follows. The detailed proof can be found in the appendix.

**Proposition 1.** *Assuming* $\frac{\exp(\mathbf{x}_i^\top \mathbf{t}_i/\tau)}{\sum_j^m \exp(\mathbf{x}_i^\top \mathbf{t}_j/\tau)} = \delta$, *we have*

$$\mathbf{x}_i^\top \mathbf{t}_k + c_1\tau \leq \mathbf{x}_i^\top \mathbf{t}_i \leq \mathbf{x}_i^\top \mathbf{t}_k + c_2\tau$$

*where $k$ denotes the nearest negative pair as $k = \arg\max_{j:j\neq i} \mathbf{x}_i^\top \mathbf{t}_j$. $c_1$ and $c_2$ are constants. $c_1 > 0$ when $\delta > 1/2$ and $c_2 > 0$ when $\delta > 1/m$.*

**Remark** Proposition 1 implies that when $\delta$ is sufficiently large, the relevant text has a higher rank than the irrelevant text, which is the key for zero-shot classification. Moreover, it shows that with the same prediction $\delta$, the absolute distance between modalities, i.e., $\|\mathbf{x}_i - \mathbf{t}_i\|_2^2 = 2 - 2\mathbf{x}_i^\top \mathbf{t}_i$, partially depends on the value of temperature, i.e., $\|\mathbf{x}_i - \mathbf{t}_i\|_2^2 \geq 2 - 2\mathbf{x}_i^\top \mathbf{t}_k - 2c_2\tau$. The analysis indicates that optimizing cross entropy loss with a small $\tau$ will not pull the text and vision space together.

Some recent work empirically observed that inter-modal distribution is significantly different from the intra-modal distribution in CLIP [18, 33], which confirms our analysis. Consequently, the proxies of class names from the text space may not be sufficient to capture the distribution of vision space, which results in the degenerated zero-shot performance. In lieu of using the text proxy directly for zero-shot visual categorization, we propose to obtain the class proxy in vision space as follows.

## 3.2 Intra-Modal Proxy Learning

With ground-truth labels $\{y_i\}$, the standard supervised learning in vision space can be written as

$$\min_W \sum_i -\log\left(\frac{\exp(\mathbf{x}_i^\top \mathbf{w}_{y_i}/\tau_I)}{\sum_j^C \exp(\mathbf{x}_i^\top \mathbf{w}_j/\tau_I)}\right) \tag{1}$$

where $\mathbf{w}_j$ is the learnable proxy for the $j$-th class and $W = [\mathbf{w}_1, \dots, \mathbf{w}_C] \in \mathcal{R}^{d\times C}$. $\tau_I$ is the temperature for the optimization with vision data. With an equivalent form, it is easy to show that $\mathbf{w}_j$ is in the vision space spanned by $\{\mathbf{x}_i\}$.

**Proposition 2.** *Let $W^*$ be the optimal solution to*

$$\min_W \sum_i -\log\left(\frac{\exp(-\|\mathbf{x}_i - \mathbf{w}_{y_i}\|_2^2/(2\tau_I))}{\sum_j^C \exp(-\|\mathbf{x}_i - \mathbf{w}_j\|_2^2/(2\tau_I))}\right)$$

*which is equivalent to Eqn. 1 if $\mathbf{w}_j$ as well as $\mathbf{x}_i$ has a unit norm. Then, we have*

$$\forall j, \quad \mathbf{w}_j^* = \Pi_{\|\mathbf{w}\|_2=1} \frac{\sum_{i:y_i=j}(1 - p_{i,j})\mathbf{x}_i - \sum_{k:y_k\neq j} p_{k,j}\mathbf{x}_k}{\sum_{i:y_i=j}(1 - p_{i,j}) - \sum_{k:y_k\neq j} p_{k,j}}$$

*where $p_{i,j} = \frac{\exp(-\|\mathbf{x}_i - \mathbf{w}_j^*\|_2^2/(2\tau_I))}{\sum_k^C \exp(-\|\mathbf{x}_i - \mathbf{w}_k^*\|_2^2/(2\tau_I))}$ and $\Pi_{\|\mathbf{w}\|_2=1}$ projects the vector to be with a unit norm.*

*Proof.* It is directly from K.K.T condition [3]. □

**Remark** Proposition 2 demonstrates that the optimal proxy for vision categorization should be in the vision space. While proxies from the text space can work as the substitute for vision proxies in CLIP, the distance between the text and vision space (i.e., the modality gap) as illustrated in Proposition 1 will degenerate the performance for vision tasks.

To further demonstrate the challenge from the gap between modalities and simplify the analysis, we decompose the proxy of class names as

$$\mathbf{z}_j = \sqrt{a}\mathbf{z}_j^x + \sqrt{1-a}\mathbf{z}_j^\perp$$

where $\mathbf{z}_j^x$ is from the vision space spanned by $\{\mathbf{x}_i\}$ and $\mathbf{z}_j^\perp$ shows the component from the orthogonal subspace such that $\mathbf{z}_j^{x\top}\mathbf{z}_j^\perp = 0$. Both $\mathbf{z}_j^x$ and $\mathbf{z}_j^\perp$ have the unit norm. $\mathbf{z}_j^\perp$ can be considered as encoding the specific information for the text modality. With the decomposition, we can observe that it is possible to recover the optimal prediction if the vision space is covered by the text space as follows.

**Proposition 3.** *Let $p_{i,j}$ and $p'_{i,j}$ denote the prediction probability obtained with vision proxies $\{\mathbf{w}_j^*\}$ and text proxies $\{\mathbf{z}_j\}$, respectively. $\tau_T$ and $\tau_I$ denote the temperature in CLIP and that in vision proxy learning, respectively. If $\mathbf{z}_j^x = \mathbf{w}_j^*$ and $0 < a < 1$, we have $p'_{i,j} = p_{i,j}$ when $\tau_I = \tau_T/\sqrt{a}$.*

**Remark** Proposition 3 shows that if the text space contains the whole vision space, the intra-modal prediction can be recovered from the cross-modal prediction using a larger temperature as $\tau_I = \tau_T/\sqrt{a}$, where $a$ measures the overlap between modalities.

However, recent observations indicate that the text and vision space learned by CLIP are distinct with a clear margin [18] and it is hard for the text space to cover the vision space. The difference between the text and vision proxy can be lower-bounded in the following theorem.

**Theorem 1.** *Let $Z$ and $W^*$ denote the text proxies and optimal vision proxies, where $Z = \sqrt{a}Z^x + \sqrt{1-a}Z^{\perp}$ and $Z^x$ consists of a rank $r$ approximation of $W^*$ as $Z^x = U_r A^{\top}$. $W^* = U\Sigma V^{\top} = \sum_i^{d'} s_i u_i v_i^{\top}$, where $d' = \min\{d, C\}$ and $s_1 \geq \cdots \geq s_{d'} \geq 0$. Then, we have*

$$\|Z - W^*\|_F^2 \geq 2C(1 - \sqrt{a}) + \sqrt{a}\sum_{i=r+1}^{d'} s_i^2$$

**Remark** Theorem 1 shows that the gap between text and vision proxy comes from two parts. The former term indicates the distance to the irrelevant text space as $1 - \sqrt{a}$. The latter one depicts the approximation loss from the low-rank overlapping between text space and vision space with $r < d'$. Due to the inherent difference between these two modalities, the distance between $Z$ and $W$ is hard to be minimized.

Therefore, to improve the zero-shot performance in the vision space, we consider to obtain the vision proxy in lieu of directly using the text proxy. The main challenge comes from the fact that the label $\{y_i\}$ is unavailable in zero-shot classification. Fortunately, the intra-modal proxy can be learned by mimicking the proxy from the other modal. Concretely, we propose to minimize the KL divergence between distributions from the text and vision proxy as

$$\min_W L(P', W) = \sum_i D_{\mathrm{KL}}(P'_i||P_i) \tag{2}$$

where $P'_i$ and $P_i$ indicate the distribution estimated by the text proxy $Z$ and the learnable vision proxy $W$, respectively.

$$p'_{i,j} = \frac{\exp(\mathbf{x}_i^{\top}\mathbf{z}_j/\tau_T)}{\sum_k^C \exp(\mathbf{x}_i^{\top}\mathbf{z}_k/\tau_T)}; \quad p_{i,j} = \frac{\exp(\mathbf{x}_i^{\top}\mathbf{w}_j/\tau_I)}{\sum_k^C \exp(\mathbf{x}_i^{\top}\mathbf{w}_k/\tau_I)}$$

Proposition 3 suggests that $\tau_I$ for $P_i$ should be larger than $\tau_T$ in $P'_i$ to calibrate the magnitude of the distribution and our ablation study confirms the analysis.

Unlike prompt learning methods [30], the problem in Eqn. 2 is defined with extracted features, which does not require the backward operation with large encoder networks for optimization. Moreover, it is convex and the optimal vision proxy can be obtained efficiently by the standard gradient descent [3].

By learning with the pseudo label predicted from the text proxy, we can recover the optimal proxy in the vision space as shown in the following theorem.

**Theorem 2.** *If simplifying the loss function as $L(P', W) = -\sum_{i,j} P'_{i,j}\log(P_{i,j})$ and assuming that it is $\mu$-strongly convex in $W$ and letting*

$$W'^* = \arg\min_W L(P', W); \quad W^* = \arg\min_W L(Y, W)$$

*where $Y_i$ is the ground-truth label distribution for $\mathbf{x}_i$, we have*

$$\|W'^* - W^*\|_F^2 \leq \frac{2}{\mu}\|P' - Y\|_F\|\log(P_{W'^*}) - \log(P_{W^*})\|_F$$

**Remark** Theorem 2 shows that the distance between the recovered vision proxy and the optimal vision proxy is bounded by the gap between pseudo labels predicted by text proxy and the ground-truth label distribution. After pre-training CLIP models on a large scale data set, it can approximate the distribution of real data, which enables learning of the intra-modal proxy.

---

**Algorithm 1** **In**tra-**M**od**a**l **P**roxy Learning (InMaP)

---

1: **Input:** Unlabeled image set $\{x_i\}$, class names $\{z_j\}$, pre-trained CLIP encoders $(f, g)$, iterations $T_w, T_p$, temperature $\tau_T, \tau_I$, threshold $\alpha$
2: Extract features $\{\mathbf{x}_i\}$ and $\{\mathbf{z}_j\}$ using pre-trained encoders $f$ and $g$, respectively.
3: Obtain pseudo labels by solving Eqn. 3 with $T_p$ iterations.
4: Refine pseudo labels with the threshold as in Eqn. 5.
5: Obtain vision proxies $W$ by solving Eqn. 2 with $T_w$ iterations.
6: **return** $y_i = \arg\max_j \mathbf{x}_i^\top \mathbf{w}_j$

---

### 3.3 Pseudo Label Refinement

According to Theorem 2, the accuracy of pseudo label is essential for recovering the appropriate vision proxy. Therefore, we develop strategies to improve the quality of pseudo labels.

First, since we have the whole test set rather than a single example for zero-shot learning, the labels obtained from the text proxy can be refined according to the geometry of the target vision data. Given the logits matrix $M \in \mathcal{R}^{n \times C}$ with $M_{i,j} = \mathbf{x}_i^\top \mathbf{z}_j$, the original pseudo label can be obtained by solving the problem

$$P' = \arg\max_P \langle P, M \rangle + \tau_T H(P) \quad s.t. \quad \forall i, \sum_j P_{i,j} = 1/n; \quad \forall i, j, P_{i,j} \geq 0$$

where $H(P)$ computes the entropy of $P$ and the obtained results should be multiplied by $n$ as the predicted labels. By incorporating a reference distribution $q \in \mathcal{R}^C$ over classes, the problem can be rewritten as

$$\max_P \langle P, M \rangle + \tau_T H(P) \quad s.t. \quad \forall i, \sum_j P_{i,j} = 1/n; \quad \forall j, \sum_i P_{i,j} = q_j; \quad \forall i, j, P_{i,j} \geq 0 \qquad (3)$$

which is known as Sinkhorn distance [8] and is an approximation of optimal transport distance. The refined pseudo label can be obtained by an efficient iterative method [8].

The last challenge is from observing the appropriate reference distribution. Without any prior knowledge, a balanced distribution $q = \mathbf{1}/C$ is a popular selection in practice. By investigating the prediction from the text proxy, it implies a distribution as $\hat{q}_j = \sum_i P'_{i,j}/n$. Then, a reference distribution can be obtained by further smoothing $\hat{q}$ as

$$q_j = \frac{\hat{q}_j^\gamma}{\sum_k \hat{q}_k^\gamma} \qquad (4)$$

where $\gamma \leq 1$ is the smoothing parameter and $\gamma = 1$ implies the original distribution from the text proxy.

Moreover, inspired by the semi-supervised learning [31], soft label with high confidence can be converted into one-hot label to eliminate the influence from irrelevant classes. Let $\alpha$ be the threshold. After obtaining the pseudo label from Eqn. 3, we have

$$\tilde{P}_i = \begin{cases} \mathbf{e}_k & k = \arg\max_j P'_{i,j}, \quad P'_{i,k} > \alpha \\ P'_i & o.w. \end{cases} \qquad (5)$$

where $\mathbf{e}_k \in \{0, 1\}^C$ is the one-hot vector and the $k$-th element is 1.

The proposed intra-modal proxy learning method (InMaP) is summarized in Alg. 1. Pre-trained CLIP models are only applied to extract features as in Step 2, which is the inevitable cost of conventional zero-shot learning. After obtaining features, the computational overhead of our proposed optimizations in Steps 3-5 is mild as shown in the analysis of running time.

## 4 Experiments

To evaluate the proposed method, we follow the common practice and conduct experiments on ImageNet [28] and 13 diverse downstream vision tasks for zero-shot transfer. Text prompts are

important for obtaining appropriate text proxies. We have the selected 7 prompts in [33] as templates for ensemble, which shows better performance than a single prompt or multiple prompts in CLIP. For the proposed method, the temperature for obtaining pseudo labels with the text proxy $\tau_T$ is set to $0.01$ as obtained by CLIP. The temperature for recovering the visual proxy $\tau_I$ is set to $0.04$ for all experiments. The intra-modal proxy is learned by standard projected gradient descent, where the initial learning rate is 10 and the number of iterations is $2,000$ for sufficient training. The learning rate will be decayed by 2 when the norm of gradient increases. Sinkhorn distance is optimized by 20 iterations for refining pseudo labels. All experiments are conducted on a single V100 GPU.

## 4.1 Ablation Study

First, we have the ablation experiments on ImageNet to demonstrate the efficacy of our method InMaP. Pre-trained ResNet-50 [13] is adopted as the vision encoder and the vanilla zero-shot method in CLIP is denoted as "Baseline".

**Temperature $\tau_I$**   $\tau_I$ is used to recover the proxy in the vision space as in Eqn. 2. According to our analysis in Proposition 3, $\tau_I$ should be larger than $\tau_T$ in CLIP to leverage the overlap between the text and vision space. Table 1 shows the result when varying $\tau_I$ in $\{0.01, 0.02, 0.03, 0.04, 0.05\}$. "Sim" computes the mean similarity between each example and its nearest proxy, depicting the gap (lower the similarly, higher the gap).

First, when $\tau_I = 0.01$, the gain is mainly from learning the vision proxy with refined labels and is already $2.59\%$ better than the baseline with text proxy. By increasing $\tau_I$, the similarity between examples and proxies also increases, which confirms our analysis in Proposition 1. With an appropriate $\tau_I$, an additional gain of $0.83\%$ can be observed with the similarity of $0.39$. According to Proposition 3, a large $\tau_I$ is crucial to recover the optimal vision proxy with the guidance from the text proxy. Thus, we fix $\tau_I = 0.04$ for the main comparison.

Table 1: Effect of $\tau_I$.

| $\tau_I$ | Acc% | Sim |
|---|---|---|
| Baseline | 60.32 | 0.26 |
| 0.01 | 62.91 | 0.31 |
| 0.02 | 63.22 | 0.32 |
| 0.03 | 63.66 | 0.35 |
| 0.04 | 63.74 | 0.39 |
| 0.05 | 63.29 | 0.43 |

Table 2: Effect of $\alpha$.

| $\alpha$ | Acc% |
|---|---|
| 1 | 63.49 |
| 0.9 | 63.53 |
| 0.7 | 63.70 |
| 0.6 | 63.74 |
| 0.5 | 63.72 |
| 0 | 63.14 |

Table 3: Effect of $\gamma$.

| $\gamma$ | Acc% |
|---|---|
| 1 | 60.95 |
| 0.5 | 62.98 |
| 0.1 | 63.75 |
| 0.05 | 63.73 |
| 0.01 | 63.74 |
| 0 | 63.74 |

Table 4: Components in InMaP.

| methods | Acc% |
|---|---|
| Baseline | 60.32 |
| InMaP$^{0.25}$ | 60.83 |
| InMaP$^{0.5}$ | 60.95 |
| Sinkhorn | 62.53 |
| InMaP | 63.74 |

**Pseudo label threshold $\alpha$**   Since the SoftMax operator can incur the over-smoothing issue for output logits, converting the soft label with high confidence to one-hot is an effective strategy in semi-supervised learning. We adopt and evaluate the performance with the threshold $\alpha$ in Eqn. 5.

Table 2 summarizes the results with different thresholds. When $\alpha = 1$, the soft label after Sinkhorn distance optimization is adopted for optimization, which already surpasses the baseline by a clear margin of $3.17\%$. When reducing $\alpha$, more soft labels will be converted to one-hot labels and the performance can be further improved by eliminating noise from the SoftMax operator. When letting $\alpha = 0$, all pseudo labels become one-hot, which amplifies the noise from low-confidence prediction and degenerates the performance. We find that our method is insensitive to the parameter and fix it as $0.6$ in the remaining experiments if not specified.

**Pseudo label smoothing $\gamma$**   While the validation set of ImageNet is well-balanced with 50 examples for each class, we conduct the experiments with different $\gamma$ in Eqn. 4 for demonstration.

Table 3 shows the accuracy on ImageNet with varying $\gamma$. $\gamma = 1$ follows the distribution from the text prediction and degenerates to the original pseudo label without Sinkhorn distance refinement. By balancing the distribution of the class assignment with a small $\gamma$, the accuracy can be increased with refined pseudo labels. When $\gamma$ is sufficiently small, the constraint in Sinkhorn distance degenerates to the balanced constraint and shows the desired performance due to the consistency with the ground-truth distribution. We will have the balanced prior distribution as $\gamma = 0$ for other data sets if not specified.

**Components in InMaP** Finally, we show the gain from two components in InMaP in Table 4. Let "InMaP$^{0.25}$" denote the proxy learning without refined labels and "InMaP$^{0.5}$" be the variant optimized by solely applying the threshold $\alpha$ to pseudo labels from the text proxy. "Sinkhorn" denotes the pseudo labels refined by Sinkhorn distance without proxy learning. We can observe that with the labels predicted from the text proxy, the recovered vision proxy of InMaP$^{0.25}$ can improve the baseline by $0.51\%$ with the reduction in the modality gap. Then, simply thresholding the prediction from the text proxy can further improve the performance by $0.12\%$ as in InMaP$^{0.5}$. On the other hand, refining the labels via Sinkhorn distance can substantially outperform the baseline by $2.21\%$. Learning with better pseudo labels, InMaP shows an additional gain of $1.21\%$ over refined labels. It confirms our analysis in Theorem 2 that the proposed intra-modal proxy learning can benefit from appropriate pseudo labels.

**Number of text prompts** An ensemble of text prompts shows the superior performance over the single prompt on obtaining text proxy for zero-shot classification [26]. We compare these two strategies on our method and summarize the accuracy on ImageNet in Table 5.

Table 5: Comparison of accuracy(%) on ImageNet with different text prompts. "Single" denotes the application of the single prompt of "a photo of a ().", while "ensemble" contains 7 prompts as suggested by [33].

| Vision encoder | Baseline_single | Baseline_ensemble | InMaP_single | InMaP_ensemble |
|---|---|---|---|---|
| ResNet-50 | 58.15 | 60.32 | 63.15 | 63.74 |
| ViT-B/16 | 66.72 | 68.75 | 71.93 | 72.55 |

First, we can observe that the ensemble improves the performance of baseline with a clear margin of $2\%$ with different vision encoders. It shows that the vanilla zero-shot transfer strategy is sensitive to the selection of text prompts and confirms the motivation of prompt learning methods. Second, our method outperforms the baseline with the single or ensemble of text prompts for text proxy, which demonstrates the effectiveness of our proposed intra-modal vision proxy. Moreover, the gap between the single and ensemble prompts shrinks on InMaP to less than $1\%$, which can help reduce the efforts of text prompt tuning.

**Running time** InMaP consists of two optimization problems, that is, pseudo label optimization in Eqn. 3 and proxy learning in Eqn. 2. The former one can be formulated as optimizing the Sinkhorn distance that can be solved efficiently by the standard iterative method in [8]. With 20 iterations, the running time is only 0.002 second, which is negligible. The latter problem is convex and defined with only extracted features. Unlike prompt learning, the expensive encoders are not included in optimization. Even with $2,000$ iterations for gradient descent, the running time is only about 30 seconds on a single GPU and the cost can be further reduced to 6 seconds with FP16. In addition, many efficient methods are available for solving convex problems as in Eqn. 2 and can be explored if needed [3].

## 4.2 Comparison on ImageNet

In this subsection, we compare InMaP to state-of-the-art methods on ImageNet with different vision encoders provided by CLIP. Table 6 summarizes the results. Besides the vanilla zero-shot prediction in CLIP, TPT [30], SuS-X [33] and VisDesc [20] are included in the comparison, where TPT leverages test data as ours but for prompt learning and the other two employ external large models that are marked in grey color. Results for these methods are directly borrowed from their original papers.

First, we can observe that our method InMaP$^{0.5}$ outperforms the baseline and TPT with both ResNet-50 and ViT-B/16 as the vision encoder. Unlike the baseline and TPT that optimizes text prompts for images, we propose to reconstruct the proxy in vision space, which reduces the influence from the modality gap and obtains an appropriate intra-modal classifier as demonstrated in Theorem 2. In addition, the running time of proxy learning mainly depends on the dimension of extracted features and that with features from ViT-L/14@336 runs faster than ResNet-50, which guarantees the efficiency with different encoders. By further improving pseudo labels with Sinkhorn distance, InMaP shows the best performance among all methods and even surpasses those ones with additional large models, which confirms the effectiveness of our method. Moreover, InMaP has a steady improvement over InMaP$^{0.5}$ using different encoders. The observation indicates that label refinement

Table 6: Comparison of accuracy (%) on ImageNet with different vision encoders in CLIP. † in grey indicates the application of external large models. The overall best performance is in bold, while the best performance without any additional model is underlined. "-" denotes that the result is unavailable in their original papers.

| Vision encoder | Baseline [26] | TPT [30] | InMaP$^{0.5}$ | InMaP | VisDesc$^†$ [20] | SuS-X-LC$^†$ [33] | SuS-X-SD$^†$ [33] |
|---|---|---|---|---|---|---|---|
| ResNet-50 | 60.32 | 60.74 | 60.95 | **63.74** | 59.68 | 61.89 | 61.84 |
| ViT-B/32 | 63.77 | - | 64.82 | **67.29** | 62.97 | 64.73 | 64.71 |
| ViT-B/16 | 68.75 | 68.98 | 70.17 | **72.55** | 68.03 | 70.00 | 69.88 |
| ViT-L/14 | 75.96 | - | 77.14 | **79.29** | 75.00 | - | - |
| ViT-L/14@336 | 77.02 | - | 78.27 | **80.21** | 76.16 | - | - |

is complementary to the intra-modal proxy learning, which is consistent with Theorem 2. Finally, with the largest backbone provided by CLIP, InMaP achieves $80.21\%$ accuracy on ImageNet, which shows the potential of large pre-trained vision-language models.

After the optimization with the target data set, we demonstrate the performance of InMaP when the target data arrives in a streaming manner but an unlabeled set of relevant data is available. Concretely, the vision proxy is constructed from the unlabeled training set in lieu of the target validation set while the learned proxy is evaluated on the validation set. Table 7 shows the results of our method learned with unlabeled training set ("InMaP$_{\text{train}}$") or validation set ("InMaP$_{\text{val}}$") of ImageNet. For InMaP$_{\text{train}}$, $\tau_I$ and $\alpha$ are reduced to $0.03$ and $0.4$, respectively, while other parameters remain unchanged.

Table 7: Comparison of accuracy (%) on ImageNet. For InMaP, "train" and "val" denote that the vision proxy is obtained from the unlabeled training set and validation set, respectively.

| Vision encoder | ResNet-50 | ViT-B/32 | ViT-B/16 | ViT-L/14 | ViT-L/14@336 |
|---|---|---|---|---|---|
| InMaP$^{0.5}_{\text{val}}$ | 60.95 | 64.82 | 70.17 | 77.14 | 78.27 |
| InMaP$_{\text{val}}$ | 63.74 | 67.29 | 72.55 | 79.29 | 80.21 |
| InMaP$^{0.5}_{\text{train}}$ | 61.38 | 65.12 | 70.37 | 77.69 | 78.68 |
| InMaP$_{\text{train}}$ | 63.98 | 67.37 | 72.59 | 79.21 | 80.13 |

We can observe that even without the target data for recovering the vision proxy, an appropriate unlabeled data set is sufficient to obtain effective proxy for zero-shot classification. By thresholding the original pseudo labels, InMaP$^{0.5}_{\text{train}}$ outperforms InMaP$^{0.5}_{\text{val}}$ over all vision encoders, where the benefit may come from the large size of the training set. Then, if refining pseudo labels with Sinkhorn distance, the performance of proxies obtained with different data becomes similar and is better than the variant without optimized pseudo labels, which confirms the efficacy of our strategy for improving labels. Finally, this experiment demonstrates that the proposed method is applicable for different scenarios in real-world applications.

### 4.3 Comparison on 13 Downstream Tasks

Then, we evaluate the proposed method on 13 diverse downstream tasks, including Aircraft [19], Caltech101 [10], Stanford Cars [16], CIFAR-10 [17], CIFAR-100 [17], CUB200-2011 [35], Describable Textures Dataset (DTD) [7], EuroSAT [14], Flowers [21], Food101 [2], Oxford-IIIT Pet (Pets) [24], Sun397 [39], and UCF101 [32]. These data sets cover a large range of classification tasks such as fine-grained visual categorization, texture recognition, scene categorization, classification with low-resolution images, etc. The same parameters used for ImageNet are applied for most of them except certain parameters for refining labels. Concretely, Caltech and Flowers can benefit from the appropriate reference distribution for generating pseudo labels and we set $\gamma$ in Sinkhorn distance as $\gamma = 0.9$ for ResNet and $\gamma = \{1, 0.5\}$ for ViT, respectively for the two data sets. Moreover, the threshold $\alpha$ is reduced to $0.3$ on EuroSAT. Other parameters remain unchanged.

Table 8 summarizes the comparison. First, we observe that with the same parameters for all data sets, InMaP$^{0.5}$ shows the better performance than the baseline consistently and obtains $1\%$ improvement on average by ResNet-50. It demonstrates that recovering the proxy in vision space can benefit different tasks for zero-shot transfer with pre-trained CLIP models. Second, by incorporating refined labels from optimizing Sinkhorn distance, InMaP can further improve InMaP$^{0.5}$ by more than $4\%$ on average. The superior performance confirms that appropriate labels can facilitate the intra-modal

Table 8: Comparison of accuracy (%) on 13 diverse downstream tasks with ResNet-50 and ViT-B/16. † in grey indicates the application of external large models. The overall best performance is in bold, while the best performance without any additional model is underlined. "-" denotes that the result is unavailable in their original papers.

| | Aircraft | Caltech | Cars | Cifar10 | Cifar100 | CUB | DTD | EuroSAT | Flowers | Food | Pets | SUN | UCF101 | Avg. |
|---|---|---|---|---|---|---|---|---|---|---|---|---|---|---|
| *ResNet-50:* | | | | | | | | | | | | | | |
| Baseline | 16.62 | 86.00 | 56.31 | 73.15 | 40.60 | 41.37 | 41.13 | 26.90 | 63.13 | 74.10 | 81.85 | 59.25 | 55.56 | 55.07 |
| TPT | 17.58 | 87.02 | 58.46 | - | - | - | 40.84 | 28.33 | 62.69 | 74.88 | 84.49 | 61.46 | 60.82 | - |
| InMaP$^{0.5}$ | 16.95 | 86.41 | 58.11 | 74.23 | 40.72 | 41.82 | 42.14 | 27.52 | 66.29 | 74.95 | 82.94 | 59.95 | 57.49 | 56.12 |
| InMaP | 18.96 | 86.73 | 63.30 | 78.84 | 49.26 | 49.17 | 44.86 | 35.88 | 66.68 | 78.36 | 87.93 | 63.82 | 63.36 | 60.55 |
| VisDesc† | 16.26 | 88.11 | 54.76 | 73.22 | 39.69 | 48.31 | 41.96 | 37.60 | 65.37 | 76.80 | 82.39 | 59.84 | 58.47 | 57.14 |
| SuS-X-LC† | 21.09 | 89.57 | 57.17 | 74.95 | 44.48 | 48.86 | 49.23 | 44.23 | 67.07 | 77.62 | 86.59 | 63.01 | 61.49 | 60.41 |
| SuS-X-SD† | 19.47 | 89.53 | 57.27 | 74.69 | 44.63 | 49.12 | 50.59 | 45.57 | 67.72 | 77.58 | 85.34 | 62.95 | 61.54 | 60.46 |
| *ViT-B/16:* | | | | | | | | | | | | | | |
| Baseline | 23.13 | 93.51 | 66.29 | 91.09 | 67.29 | 49.36 | 45.09 | 50.17 | 67.64 | 84.43 | 87.00 | 65.68 | 65.21 | 65.84 |
| TPT | 24.78 | 94.16 | 66.87 | - | - | - | 47.75 | 42.44 | 68.98 | 84.67 | 87.79 | 65.50 | 68.04 | - |
| InMaP$^{0.5}$ | 23.55 | 93.59 | 67.96 | 92.48 | 68.37 | 51.54 | 46.28 | 53.64 | 69.64 | 85.76 | 88.17 | 67.18 | 67.25 | 67.34 |
| InMaP | 28.35 | 93.59 | 73.00 | 93.27 | 72.51 | 57.87 | 50.77 | 63.98 | 71.28 | 87.76 | 92.94 | 70.85 | 72.06 | 71.40 |
| VisDesc† | - | - | - | - | - | 57.75 | 45.59 | 48.82 | - | 88.50 | 86.92 | - | - | - |
| SuS-X-LC† | 30.51 | 93.91 | 65.90 | 90.94 | 68.66 | 56.96 | 55.32 | 58.06 | 73.08 | 86.08 | 91.58 | 67.85 | 66.72 | 69.66 |
| SuS-X-SD† | 28.68 | 93.96 | 66.13 | 89.88 | 68.47 | 57.11 | 54.55 | 57.49 | 73.81 | 86.08 | 90.57 | 67.73 | 66.59 | 69.31 |

proxy learning to obtain better proxy reconstruction in the target space. In addition, our method is the best among all methods on 8 out of 13 data sets with ResNet-50, which shows that leveraging unlabeled target data can be more effective than external models for zero-shot learning. Moreover, the same parameters are shared by different tasks for InMaP, while fine-tuning them by a validation set for each data set can further improve the performance.

With ViT-B as the vision backbone, the accuracy of the baseline increases by $10\%$, which shows the potential of large models. The proposed method can also benefit from the large models and achieve the accuracy of $71.4\%$ on average, which is more than $5\%$ better than the baseline. Compared with SuS-X with external models, InMaP outperforms it by $1.74\%$ that confirms the effectiveness of intra-modal proxy learning. Moreover, for the data set with a high zero-shot accuracy as Caltech, the predicted distribution can approximate the ground-truth well and the label refinement with Sinkhorn distance can be skipped by letting $\gamma = 1$. Finally, both InMaP$^{0.5}$ and InMaP improves over the baseline on diverse tasks with ResNet and ViT, which illustrates that the proposed method is applicable for different vision encoders.

## 5  Conclusion

In this work, we focus on improving the zero-shot transfer with CLIP. First, our theoretical analysis indicates that the modality gap between the text and vision space obtained by CLIP will be preserved, which can degenerate the performance of zero-shot transfer. To mitigate the challenge, we propose to recover the proxy of each class in the vision space with the help from the text proxy. Concretely, by leveraging the unlabeled target data and refining the prediction from the text proxy, we can obtain the vision proxy via learning with unlabeled data and the corresponding pseudo label. Experiments on diverse data sets demonstrate that our method can improve the zero-shot accuracy consistently with different vision encoders. Exploring our method with other pre-trained models rather than CLIP can be our future work. Besides, recent work [33] shows that other large pre-trained models, i.e., GPT-3, can help CLIP for zero-shot prediction. Combining our method with existing large models is also an interesting future direction. Finally, our analysis for the efficacy of the text proxy is from the perspective of approximating the supervised learning for images as in Eqn. 1. Text information can be complementary to the knowledge in images [36] and investigating the potential benefit of text for vision tasks can help understand vision-language models better.

## Limitations

While our analysis is for the modality gap in contrastive pre-training with the architecture of dual encoders, it cannot be applied directly for other pre-training paradigms with different architectures. For example, BEiT-3 [37] has masked data modeling for pre-training with the shared backbone for different modalities and CoCa [40] has encoder-decoder captioning for pre-training. The analysis for the modality gap in these methods has been less investigated and can be inspired by our results as a future direction.

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

# A Theoretical Analysis

## A.1 Proof of Proposition 1

*Proof.* According to the definition, we have

$$(1 - \delta) \exp(\mathbf{x}_i^\top \mathbf{t}_i / \tau) = \delta \sum_{j:j \neq i}^{m} \exp(\mathbf{x}_i^\top \mathbf{t}_j / \tau)$$

Hence

$$\exp(\mathbf{x}_i^\top \mathbf{t}_i / \tau) = \frac{\delta}{1 - \delta} \sum_{j:j \neq i}^{m} \exp(\mathbf{x}_i^\top \mathbf{t}_j / \tau) \leq \frac{\delta(m-1)}{1 - \delta} \exp(\mathbf{x}_i^\top \mathbf{t}_k / \tau)$$

$$\exp(\mathbf{x}_i^\top \mathbf{t}_i / \tau) \geq \frac{\delta}{1 - \delta} \exp(\mathbf{x}_i^\top \mathbf{t}_k / \tau)$$

where $k = \arg\max_{j:j \neq i} \mathbf{x}_i^\top \mathbf{t}_j$. Since logarithm function is monotone, the similarity between positive pair can be bounded by

$$\mathbf{x}_i^\top \mathbf{t}_k + c_1 \tau \leq \mathbf{x}_i^\top \mathbf{t}_i \leq \mathbf{x}_i^\top \mathbf{t}_k + c_2 \tau$$

where $c_1 = \log(\frac{\delta}{1-\delta})$ and $c_2 = \log(\frac{\delta(m-1)}{1-\delta})$. $\square$

## A.2 Proof of Proposition 3

*Proof.* According to the definition, we have

$$p'_{i,j} = \frac{\exp(\mathbf{x}_i^\top \mathbf{z}_j / \tau_T)}{\sum_k \exp(\mathbf{x}_i^\top \mathbf{z}_k / \tau_T)} = \frac{\exp(\sqrt{a}\mathbf{x}_i^\top \mathbf{z}_j^x / \tau_T)}{\sum_k \exp(\sqrt{a}\mathbf{x}_i^\top \mathbf{z}_k^x / \tau_T)} = \frac{\exp(\sqrt{a}\mathbf{x}_i^\top \mathbf{w}_j^* / \tau_T)}{\sum_k \exp(\sqrt{a}\mathbf{x}_i^\top \mathbf{w}_j^* / \tau_T)}$$

If $\sqrt{a}/\tau_T = 1/\tau_I$, the predictions are equivalent. $\square$

## A.3 Proof of Theorem 1

*Proof.* According to the definition, we have

$$\|Z - W^*\|_F^2 = 2C - 2\sqrt{a}\langle Z^x, W^* \rangle = 2C(1 - \sqrt{a}) + \sqrt{a}\|Z^x - W^*\|_F^2 \tag{6}$$

To bound the similarity between $Z^x$ and $W^*$, we assume $Z^x = U_r A^\top$ where $U_r \in \mathcal{R}^{d \times r}$ is a subset of $U \in \mathcal{R}^{d \times d'}$. By solving the problem $\min_A \|U_r A - W^*\|_F^2$, we have $Z^x = U_r A = U_r(U_r^\top U_r)^{-1} U_r^\top W^* = U_r U_r^\top W^*$ that is projecting $W^*$ to the subspace spanned by $U_r$. Since $U_r$ is a subset from $U$, we have $U_r U_r^\top = \sum_i^{d'} r_i u_i u_i^\top$ with $\forall i, r_i \in \{0, 1\}$ and $\sum_i r_i = r$. Then, we have

$$\|U_r U_r^\top W^* - W^*\|_F^2 = \|\sum_i (r_i - 1) s_i u_i v_i^\top\|_F^2 = \sum_i (r_i - 1)^2 s_i^2$$

Therefore

$$\|Z^x - W^*\|_F^2 \geq \|U_r U_r^\top W^* - W^*\|_F^2 \geq \sum_{i=r+1}^{d'} s_i^2$$

where the last inequality is from keeping the largest singular values for minimizing the approximation loss. The target result is obtained by taking this inequality to Eqn. 6. $\square$

## A.4 Proof of Theorem 2

*Proof.* First, we note that $L(P, W)$ is a convex function in $W$. We assume that it is $\mu$-strongly convex in $W$ such that for the arbitrary $(W_1, W_2)$, we have

$$L(P, W_1) \geq L(P, W_2) + \langle \nabla_{W_2} L(P, W_2), W_1 - W_2 \rangle + \frac{\mu}{2}\|W_1 - W_2\|_F^2$$

According to the optimality of $W'^*$, we have

$$L(P', W'^*) - L(Y, W^*) \leq L(P', W^*) - L(Y, W^*) = \langle P' - Y, -\log(P_{W^*}) \rangle \qquad (7)$$

The lower-bound can be obtained from the strongly convexity of $W$ as

$$L(P', W'^*) - L(Y, W^*) = L(Y, W'^*) - L(Y, W^*) + \langle P' - Y, -\log(P_{W'^*}) \rangle$$
$$\geq \frac{\mu}{2} \|W'^* - W^*\|_F^2 + \langle P' - Y, -\log(P_{W'^*}) \rangle \qquad (8)$$

Combining inequalities in Eqns. 7 and 8, we have

$$\|W'^* - W^*\|_F^2 \leq \frac{2}{\mu} \langle P' - Y, \log(P_{W'^*}) - \log(P_{W^*}) \rangle \leq \frac{2}{\mu} \|P' - Y\|_F \| \log(P_{W'^*}) - \log(P_{W^*}) \|_F$$

$\square$

