# OpenReview forum: "Intra-Modal Proxy Learning for Zero-Shot Visual Categorization with CLIP"
_NeurIPS.cc/2023/Conference — NeurIPS 2023 poster_

### Official Review · Reviewer_DJum · 2023-06-30

**Soundness:** 3 good
**Presentation:** 2 fair
**Contribution:** 2 fair
**Rating:** 5
**Confidence:** 3

**Summary:**

Authors provide a series of detailed theories to support the conclusion that the inter-modal distance of the CLIP model is large and the gap can be reduced by introducing proxy features in visual feature space. They also propose the intra-modal proxy learning method to improve the performance of CLIP model with only unlabeled test data. Experiments on several datasets and different network architectures validate their proposed method's effectiveness.


**Strengths:**

1. Motivation is clear and theoretical analysis is very sufficient.
2. The mehtod derived from theoretical analysis is convincing.
3. The paper is well-organized.

**Weaknesses:**

1. Temperature in CLIP paper is a learnable parameter which is initialized as 0.07 and becomes 100 after training. Is Tau_T in submission the same parameter as the temperature t in CLIP? And why set to 0.01.
2. According to Table 2, it seems alpha contributions little to the proposed method. If we ignore the step, the acc will drop from 63.74 to 63.49, not too much for a single experiment.
3. Can authors provide the results on UCF101 and eurosat, which are common datasets used in CLIP tuning methods.
4. Can authors provide the results of default prompt instead of 7.
5. About TPT results, authors copy the results from their paper, resutls of CLIP baseline is obvious higher than their bsaeline, why? If bsaeline uses ensemble prompt, please refer to Weakness(4). Using high baseline results and copied TPT results seems unfair.


**Questions:**

Please refer to Weaknesses.

**Limitations:**

No potential negative societal impact.

---

> ### Author Rebuttal · Authors · 2023-08-09
>
> R-DJum-W1-A: Yes, they are the same. 0.01 was obtained by CLIP and we did not change it. We will make it clear in revision.
>
> R-DJum-W2-A: Please kindly note that the result is obtained on ImageNet, where achieving a gain of 0.25\% is not trivial. Although the parameter can be eliminated to reduce tuning efforts, we find that it is not sensitive and the same value can be applied for most of data sets.
>
> R-DJum-W3-A: Thanks for the suggestion. The results on UCF101 and EuroSAT are summarized in Table 1 in the attached PDF and we will include them in the revision. With the additional data sets, the main conclusion over the 13 data sets remains the same, where the proposed method achieves the best average performance among all methods.
>
> R-DJum-W4-A: Most of our baselines that relies on handcrafted prompts have 7 prompts, so we keep the same setting for the fair comparison. Note that TPT aims to learn the optimal prompt where the handcrafted prompts are inapplicable. The results with a single prompt of \``a photo of a ( ).\'' on ImageNet are provided in Table 2 (i.e., denoted by ``single'') in the attached PDF. It can be observed that our method still demonstrates a consistent improvement over baseline and outperforms TPT with a clear margin of more than 2\%. Thanks for pointing out this. We will add this discussion in revision.
>
> R-DJum-W5-A: The CLIP baseline reported in our work is with 7 prompts suggested by [26], which is different from the baseline in TPT with a single prompt or an ensemble of 80 prompts. It should be noted that TPT aims to learn the optimal prompt that does not require the ensemble of prompts and is better than the ensemble. Therefore, the comparison is fair and TPT also includes the ensemble as the baseline in their original paper. Finally, we compare TPT with a single prompt in Table 2 in the attached PDF, where our method still shows a better performance by more than 2\% on ImageNet. We will add this discussion and make it clear.

---

> ### Comment · Reviewer_DJum · 2023-08-20
>
> Thanks to the author's reply and supplementary experiments, I still maintain my rating as Borderline accept.

---

### Official Review · Reviewer_fSSP · 2023-07-06

**Soundness:** 3 good
**Presentation:** 2 fair
**Contribution:** 2 fair
**Rating:** 5
**Confidence:** 3

**Summary:**

This work theoretically proves that the gap cannot be reduced sufficiently by minimizing the contrastive loss in CLIP and the optimal proxy for vision tasks resides only in the vision space. Two strategies are developed to further refine the pseudo label obtained by the text proxy to facilitate intra-modal proxy learning (InMaP). Experiments on ImageNet and extensive downstream tasks confirm the effectiveness and efficiency of the proposal.

However, I have some concerns. Please refer to the comments below.

**Strengths:**

- This paper is easy to understand.
- To mitigate the problem of the gap between the text and vision space, the authors theorize that the optimal proxy for visual tasks is solely from visual space. This is reasonable and convincing.
- Extensive experiments are performed on different datasets.

**Weaknesses:**

- For NeurIPS, the innovation or contribution of this work is limited, especially the technical solution. Proxy learning is not a new technique. Furthermore, pseudo label refinement is a combination of existing methods.

- Related work introduces few-shot transfer rand zero-shot transfer. It seems that this paper does not report the performance of few-shot transfer (this paper uses CLIP as the Baseline, and CLIP performs a few-shot transfer experiment).

- The layout of this paper is not very reasonable.

**Questions:**

- There seem to be some syntax issues, please check carefully. For example, Line 44 and Line 48.

- Please distinguish between ablation studies and hyperparameter analysis. Does this paper include hyperparameter analysis in "Alation on ImageNet"? This confuses me.

- For some comparison methods, this paper shows by lighter color, why is this? It doesn't seem to be clearly stated.

**Limitations:**

- This paper does not analyze the reasons for achieving zero-shot performance gains or provide deeper insights into the field.

- See Weaknesses and Questions.

---

> ### Author Rebuttal · Authors · 2023-08-09
>
> R-fSSP-W1-A: The main contribution of this work is to theoretically understand the modality gap in CLIP, which has also inspired our novel proposal of proxy learning. To the best of our knowledge, this is the first work to explore intra-modal proxy learning for CLIP. Meanwhile, to better recover the appropriate vision proxy as implied by our theoretical analysis, we propose a novel pseudo label refinement strategy based on Sinkhorn distance, where it is not trivial to obtain an appropriate reference distribution for Sinkhorn distance as discussed in Line 210 and 215. Thanks for pointing out this. We will make this clear in revision.
>
> R-fSSP-W2-A: As discussed in Line 32: ``labels from the target domain may not be available'' and stated in Line 39, our work focuses on zero-shot learning. Many existing methods also solely aim to improve the zero-shot performance of CLIP as discussed in the related work. Thanks for pointing out this. We will make this clear in revision.
>
> R-fSSP-W3-A: Thanks for the suggestion. We will improve the organization and make it clear.
>
> R-fSSP-Q1-A: We will thoroughly check the writing as suggested in revision.
>
> R-fSSP-Q2-A: We included the analysis of hyper-parameter in the section of ``Ablation on ImageNet'' and the results can be found in Tables 1-3. Thanks for pointing out this and we will make it clear in revision.
>
> R-fSSP-Q3-A: As mentioned in the caption of Table 5, methods with grey results denoted by $\dagger$ ``indicates the application of external large models'', which includes GPT-3 [1] and Stable Diffusion [2]. We will make it clear in revision.
>
> R-fSSP-L-A: Please kindly note that the main contribution of this work is to theoretically understand the modality gap in CLIP and derive our novel proposal of intra-modal proxy learning accordingly. The performance gain is demonstrated theoretically in Theorem 1 and 2, and empirically on ImageNet and diverse tasks in Table 5 and 6. This work can help better understand CLIP and inspire further improvement in the field. We will clarify in revision.
>
> [1] Language Models are Few-Shot Learners. NeurIPS 2020.
> [2] High-Resolution Image Synthesis with Latent Diffusion Models. CVPR 2022.

---

> > ### Comment · Reviewer_fSSP · 2023-08-15
> > **Response to Authors After Checking the Rebuttal**
> >
> > Thanks to authors' rebuttal for further clarification.
> >
> > I still have other concerns:
> >
> > i) As CLIP used large number of data for learning, which inevitably results in data discloure. As such, taking CLIP for zero-shot visual categorization is not very acceptable. I want to know how did author gurantee the real zero-shot generalization in the experiments.
> >
> > ii) As we all known, there are lots of STANDARD zero-shot leanring (ZSL) methods [r1-r6, etc.] applied into image classification/obeject recognition.  Different to CLIP-based zero-shot generalization mehtods,  STANDARD ZSL methods perform real zero-shot generalization.  As such, I strongly encourage authors take more discussions between the STANDARD ZSL and CLIP-based generalization methods.
> >
> > [r1] Zero-shot Learning with Semantic Output Codes. In NeurIPS, 2009.
> >
> > [r2] Label-Embedding for Image Classification. TPAMI, 2016.
> >
> > [r3] Zero-shot learning on semantic class prototype graph. TPAMI, 2017.
> >
> > [r4] Zero-Shot Learning: A Comprehensive Evaluation of the Good, the Bad and the Ugly. TPAMI, 2019.
> >
> > [r5] HSVA: Hierarchical Semantic-Visual Adaptation for Zero-Shot Learning. In NeurIPS, 2021.
> >
> > [r6] TransZero++: Cross Attribute-guided Transformer for Zero-Shot Learning. TPAMI, 2022.

---

> > > ### Author Response · Authors · 2023-08-16
> > >
> > > Q1: As CLIP used large number of data for learning, which inevitably results in data discloure. As such, taking CLIP for zero-shot visual categorization is not very acceptable. I want to know how did author gurantee the real zero-shot generalization in the experiments.
> > >
> > > A1: Technically, the issue of data leak can be avoided by eliminating the overlapping data from the training set. However, the analysis in CLIP shows that the overlapping with downstream tasks is very mild (e.g., 3.2\% on average) and the influence on the generalization performance is negligible on most of data sets, i.e., no more than 0.1\%. More details can be found in Section 4 titled ``Data Overlap Analysis'' in [1].
> > >
> > > Q2: As we all known, there are lots of STANDARD zero-shot leanring (ZSL) methods [r1-r6, etc.] applied into image classification/obeject recognition. Different to CLIP-based zero-shot generalization mehtods, STANDARD ZSL methods perform real zero-shot generalization. As such, I strongly encourage authors take more discussions between the STANDARD ZSL and CLIP-based generalization methods.
> > >
> > > A2: Thanks for the suggestion. We will add the discussion as follows in the revision.
> > >
> > > Zero-shot learning that aims to identify examples of novel classes without any labeled training data has been studied extensively since [r1]. Most of existing works are only able to discover new classes closely related to the training classes [r1-r6], where they share the similar attributes, and have to train an individual model for each task. On the contrary, pre-training on large-scale data with the contrastive loss aligning visual and language features makes a single CLIP model applicable for diverse downstream tasks in a straightforward way. Compared with conventional zero-shot methods, the pre-training data in CLIP may be overlapped with downstream tasks, which can result in data leak for evaluation. While the issue can be addressed by eliminating the overlapping data for training, the influence on the performance is negligible as discussed in [1].
> > >
> > > [1] Learning Transferable Visual Models From Natural Language Supervision. ICML 2021.

---

> > > > ### Comment · Reviewer_fSSP · 2023-08-21
> > > >
> > > > Thanks to authors' reply. Most of my concerns are addressed, I keep my original positive rating.

---

### Official Review · Reviewer_dkv6 · 2023-07-07

**Soundness:** 4 excellent
**Presentation:** 4 excellent
**Contribution:** 4 excellent
**Rating:** 7
**Confidence:** 4

**Summary:**

Based on the phenomenon of modality gap, this paper analyzes that the proxy of visual space is better than that of text space, and explains this theoretically. This paper designs a test-time adaptation method for unlabeled test data

**Strengths:**

1.  The influence of the modality gap on image classification is analyzed from the perspective of the proxy, and the theoretical results are amazing, which is of great significance for CLIP community research.
2. The results of the experiment are very significant, and I think this article touches on the essence of CLIP.

**Weaknesses:**

Some analysis of validity for the textual model is lacking.

**Questions:**

I once thought about the role of textual models in downstream image classification.   CoOp and other works try to optimize the textual model and get success, but from the perspective of this paper, textual models seem to play no key role in downstream tasks just as an initialization.   What is the author's opinion?

**Limitations:**

The author has not provided any limitations of this article, so I hope you can add some analysis.

---

> ### Author Rebuttal · Authors · 2023-08-09
>
> R-dkv6-W-A: Thanks for this great suggestion. With the contrastive pre-training, the textual feature will be aligned with the vision feature to enable zero-shot learning. It means that the similarity between the image feature $x_i$ and its corresponding text $t_i$ as in Proposition~1 can be lower bounded as $x_i^\top t_i\geq x_i^\top t_k + \log(\delta/(1-\delta))\tau$ for $k\ne i$. It implies that when $\delta$ is sufficiently large, the relevant text has a higher rank than the irrelevant text, which is the key for zero-shot classification. We will add this analysis in revision.
>
> R-dkv6-Q-A: Textural model provides an applicable classifier only with class names, which is a great property for zero-shot learning. However, when labeled data is available as in few-shot learning, some recent work shows that fine-tuning the classifier initialized by the textural encoder can achieve the comparable or even better performance than CoOp, i.e., Table 7 and Fig. 14 in [1].
>
> However, it should be noted that compared with the vision encoder, the textual encoder contains potential knowledge from text information that may not be covered by the vision data, for which the additional value as the regularization can be explored in the future work. We will add this discussion in the revision.
>
>
> R-dkv6-L-A: Thank you for the suggestion. We will add the following discussion.
>
> While our analysis is for the modality gap in contrastive pre-training with the architecture of dual encoders, it cannot be applied directly for other pre-training paradigms with different architectures. For example, BEiT-3 [2] has masked data modeling for pre-training with the shared backbone for different modalities and CoCa [3] has encoder-decoder captioning for pre-training. The analysis for the modality gap in these methods has been less investigated and can be inspired by our results as a future direction.
>
> [1] Robust Fine-tuning of Zero-Shot Models. CVPR 2022.
> [2] Image as a Foreign Language: BEIT Pretraining for All Vision and Vision-Language Tasks. CVPR 2023.
> [3] CoCa: Contrastive Captioners are Image-Text Foundation Models. TMLR 2022.

---

### Official Review · Reviewer_eKBJ · 2023-07-08

**Soundness:** 4 excellent
**Presentation:** 3 good
**Contribution:** 3 good
**Rating:** 6
**Confidence:** 2

**Summary:**

The paper aims to improve the zero-shot performance of CLIP, i.e.: the setting in which given a test image, we compute its visual embedding, computes its similarity to a pre-computer set of text query embeddings where each text query corresponds to a different class and finally output the classification results as class corresponding the most similar text query (in CLIP's joint embedding space).

The paper argues that zero-shot misclassifications are due to the visual-textual modality gap, i.e. the inherent gap in CLIP's embeddings space between embeddings of image and embeddings of their corresponding text queries. The paper argues (and theoretically proves) that this gap cannot be bridged by optimising on the text query and that the optimal proxy lies within the vision space. The paper proposes a novel algorithm that recovers vision proxies for text queries from unlabelled image test data. The paper demonstrates the effectiveness of the method on 12 downstream vision tasks.

**Strengths:**

* The paper addresses a key limitation of CLIP (and other vision-language models trained using contrastive loss) - the modality gap. The modality gap is an inherent fundamental "built-in" limitation of CLIP which hinders its performance. Tackling such a key problem in a wide spread and popular model (or model family) is of great important).
* The paper provides both an in-depth theoretical discussion as well as a thorough experimental evaluation demonstrating the effectiveness of the proposed method.

**Weaknesses:**

Nothing in particular that I can point to.

**Questions:**

Nothing in particular.

**Limitations:**

The authors did not addressed the limitations of their work.

---

> ### Author Rebuttal · Authors · 2023-08-09
>
> R-eKBJ-L-A: Thank you for the suggestion. We will add the following discussion.
>
>
> While our analysis is for the modality gap in contrastive pre-training with the architecture of dual encoders, it cannot be applied directly for other pre-training paradigms with different architectures. For example, BEiT-3 [1] has masked data modeling for pre-training with the shared backbone for different modalities and CoCa [2] has encoder-decoder captioning for pre-training. The analysis for the modality gap in these methods has been less investigated and can be inspired by our results as a future direction.
>
> [1] Image as a Foreign Language: BEIT Pretraining for All Vision and Vision-Language Tasks. CVPR 2023.
> [2] CoCa: Contrastive Captioners are Image-Text Foundation Models. TMLR 2022.

---

### Author Rebuttal · Authors · 2023-08-09

We would like to thank all reviewers for their valuable time and insightful comments. We will answer (A) the weaknesses (W), questions (Q) and limitations (L) for each reviewer. The experiments suggested by Reviewer DJum can be found in the attached PDF.

---

### Decision · Program_Chairs · 2023-09-21

**Decision:**

Accept (poster)

**Comment:**

This paper received accept, weak accept and borderline accept (2x) recommendations. There is a rebuttal.
Reviewer dkv6 requires some analysis of validity for the textual model and the authors provide an explanation in the rebuttal. Reviewer fSSP has various concerns around innovation / contribution and evaluation. Their concerns are mainly addressed in the rebuttal and they remain their borderline accept rating. Similarly, Reviewer Djum remains their rating.
The AC agrees with the reviewers' opinion towards accepting this work. The authors are strongly encouraged to include the additional feedback into the paper, as well as including the results from the rebuttal in the camera-ready version.